# Electron-Beam Deposition of Aluminum Nitride and Oxide Ceramic Coatings for Microelectronic Devices

**Yury G. Yushkov [1], Efim M. Oks [1,2], Andrey V. Tyunkov [1], Alexey Yu Yushenko [3] and Denis B. Zolotukhin [1,***

1 Department of Physics, Tomsk State University of Control Systems and Radioelectronics, 634050 Tomsk, Russia; yushkovyu@mail.ru (Y.G.Y.); oks@fet.tusur.ru (E.M.O.); tyunkov84@mail.ru (A.V.T.)
2 Laboratory of Electron Sources, Institute of High Current Electronics SB RAS, 634055 Tomsk, Russia
3 Department of Electronic Devices, Semiconductor Research Institute (NIIPP), 634034 Tomsk, Russia; niipp@niipp.ru
* Correspondence: denis.b.zolotukhin@tusur.ru; Tel.: +7-3822-41-47-12

**Abstract:** This work presents the results of the coating deposition by electron-beam evaporation of aluminum nitride and aluminum oxide targets in nitrogen and oxygen atmospheres in the forevacuum range (5–30 Pa). The method we employed is a combination of the electron-beam and plasma methods, since in the mentioned pressure range, the electron beam creates plasma that essentially changes the interaction picture of both the electron beam with the ceramic target and the flux of evaporated material with a substrate. We show a possibility of depositing such coatings on monolithic microwave integrated circuits passivated by $Si_3N_4$ dielectric.

**Keywords:** aluminum oxide ceramic; electron-beam evaporation; fore-vacuum pressure range; multicomponent beam plasma; dielectric coating; microelectronics



## 1. Introduction

The development of modern microelectronics that seeks improved reliability and miniaturizing of devices and their components is closely related to the development and research on new materials and their properties. It includes new ceramic composite material that can sustain high temperatures, while possessing excellent dielectric properties and being able to operate in aggressive media [1]. Aluminum nitride (AlN) has a high thermal conductivity that improves the heat sinking from microelectronic elements and devices and, hence, raises the power dissipation limit [2]. Aluminum oxide ($Al_2O_3$) possesses excellent dielectric properties, is chemically neutral and has good tribological characteristics [3]. It has become possible to deposit coatings based on these materials on the surface of microelectronic devices thanks to the use of forevacuum plasma electron sources, which are capable of creating an electron beam with a power density of 50 $kW/cm^2$ sufficient for efficient evaporation of any ceramic target at high gas pressures (1–100 Pa) [4]. In this method, it is possible to prevent excessive heating of microelectronic devices, since the heating source is point-like and the surface being deposited can be distanced from the evaporated object. This feature is a basis for classifying the proposed method of depositing ceramic coatings among low-temperature methods. When propagating in the gas of the fore-vacuum pressure (1–100 Pa), the electron beam generates plasma that eliminates the problem of the electric charge accumulation on a dielectric target and, additionally, improves the chemical reactivity of gas and preserves the surface stoichiometry, which is especially important for the deposition of nitride layers. Electron-beam evaporation of ceramics makes it possible to reach coating deposition rates up to tens of μm/min, making it a real alternative to existing methods [5].

Electron-beam evaporation of ceramics in chemically active gas media allows the properties and parameters of created coatings to be varied in a broad range by changing specific and integral parameters of the electron beam: the electron beam energy, current,

current density, pressure and a type of gas, electric potential of the irradiated surface. As far as the problems of microelectronics are concerned, when creating heat-conductive ceramic coatings, it is important to study the conditions and determine the optimal parameters of electron-beam evaporation that provide for a high degree of the coating homogeneity, the required thickness within a reasonable time, high adhesion properties on a semiconductor (silicon, etc.) wafer; coefficient of linear expansion, close to silicon and other semiconductors; low values of the relative dielectric permittivity, which is important for microwave electronics. As shown previously [6], the best combination of dielectric and heat-conductive properties can be achieved using nitride or aluminum oxide coatings, since these coatings are suitable for many applications in microelectronics [7,8]. The goal of this work was to demonstrate the compatibility of depositing AlN- and $Al_2O_3$-based coatings by an electron-beam in fore-vacuum with the technology of fabricating monolithic integral circuits (MIC). It was assumed that the heat-insulating layer of $Al_2O_3$ would reduce the heat spread to the microchip, while the heat-conductive AlN layer would distribute the penetrated heat throughout the entire surface of the chip, thus alleviating the local heat load [2,3,9].

## 2. Materials and Methods

The experimental schematic diagram is shown in Figure 1. Experiments were carried out on an installation equipped (Laboratory of Plasma Electron Sources, Tomsk, Russia) with a fore-vacuum plasma-cathode electron source based on the electron emission from a hollow-cathode discharge plasma, operating in a continuous mode in the pressure range of 1–100 Pa. At an accelerating voltage of 20 kV and a maximum discharge current of 1.5 A, the electron beam current reached 450 mA and the maximum beam power density could reach up to 50 $kW/cm^2$ with a diameter of 0.5–10 mm. The arcing due to dielectric films deposition on electrodes of the electron source was eliminated thanks to a special construction of the electrodes system as well as the deflection of the e-beam focal point from the axis of the symmetry of the electron source. The surface temperature of the deposited sample during the coating process was monitored by an optical pyrometer Raytek of Marathon MM series (Raytek, Santa Cruz, CA, USA). A more detailed description of the experimental setup is given in [10]. The dielectric coatings were deposited in several stages, during each stage the coating thickness increased (thickness of each layer was 500 nm). The electron beam power density was no more than 1 $kW/cm^2$.

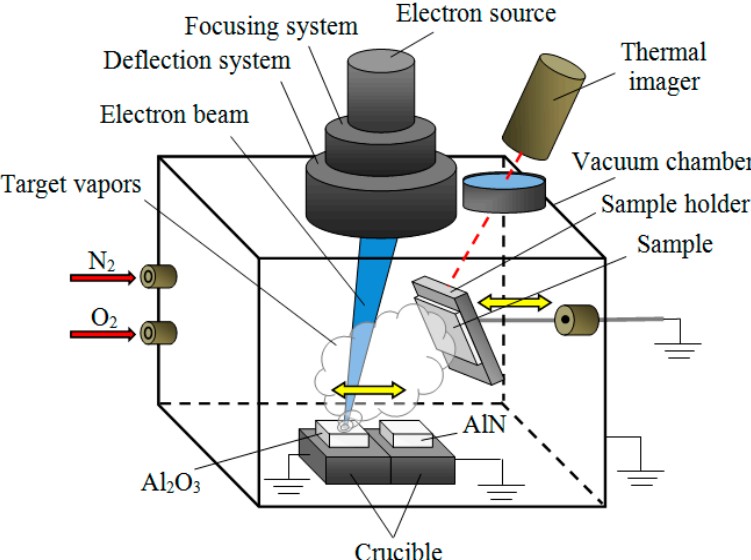

**Figure 1.** Experimental setup for coatings deposition on a microelectronic device.

The study of compatibility between the AlN- and $Al_2O_3$-based coating deposition technique and monolithic microwave integrated circuits (MIC) was carried out using a

double balanced mixer MM608 of NIIPP Ltd (NIIPP Ltd, Tomsk, Russia). manufactured for the frequency range 4–18 GHz. The MM608 mixer was produced by NIIPP [11] using the basic MIC technology on vertical Schottky diodes. The MM608 mixer is built on a ring circuit and includes two Marchand baluns, four diodes, and a low frequency filter based on a spiral inductor and MDM (Metal-Dielectric-Metal) capacitors. The MDM-capacitors and MIC passivation are made using $Si_3N_4$ dielectric. The MIC mixer dimensions are $1.5 \times 1.3 \times 0.1$ mm$^3$. Considering the passivation of the active elements of the MIC, it was assumed that the main factor of the possible degradation of its active elements (Schottky diodes) could be the high temperature during depositing the dielectric coatings. After deposition of each couple of layers ($Al_2O_3$ and AlN), the main parameters of the mixer were measured: conversion loss and LO (Local Oscillator) to RF (Radio Frequency) isolation.

The dielectric coating on the MIC was deposited in the following way. First, the electron beam evaporated the aluminum oxide target for one minute in the oxygen atmosphere at a pressure of 8 Pa. Then nitrogen gas was injected into the chamber at 8 Pa, and using the magnetic deflection system the beam was deflected to the aluminum nitride ceramics, and evaporated the target of aluminum nitride ceramics. After that, the beam focal point was returned to the aluminum oxide target again to evaporate it. Aluminum oxide target was evaporated in oxygen, aluminum nitride target—in nitrogen. In this manner, the formation of alternating layers of different ceramic materials has been performed on the substrate. Due to the facts that (1) the MIC is fragile and (2) it has a complex surface, the thickness of the coating deposited directly on the MIC is hard to be determined. Therefore, in order to determine the thickness of the coatings obtained, a "witness" of silicon was positioned near the chip. Figure 2 shows a picture that illustrates the process of the coating depositing on the chip surface by electron-beam evaporation of ceramics.

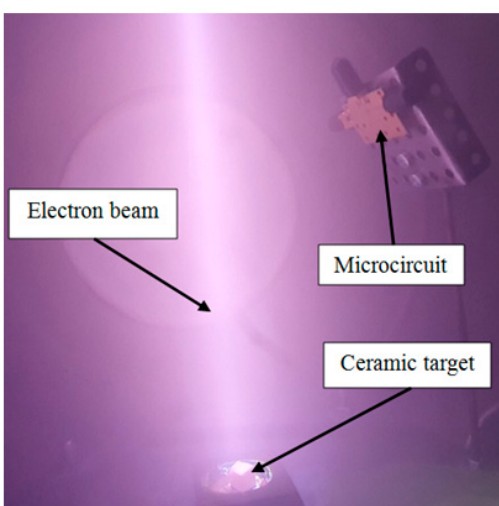

**Figure 2.** The process of electron-beam coating deposition on a MIC sample.

To avoid the destruction of the MIC sample, we experimentally selected the optimal temperature for coating deposition, which was less than 200 °C. The choice of temperature was made according to the MIC datasheet operational requirements and to our preliminary experiments where we have found that at the MIC sample temperatures higher than 200 °C the passivating layer on its surface started to degrade. To ensure the required temperature, the sample was located at a distance of 10 cm from the ceramic being evaporated or at a distance of 2 cm with the density of the electron beam power of up to 300 W/cm$^2$, see Figure 3.

Elemental composition of the deposited coatings was examined used a Hitachi S3400N scanning electron microscope equipped (Hitachi, Tokyo, Japan) with a BrukerX'Flash 5010 energy dispersive analyzer (Billerica, MA, USA).

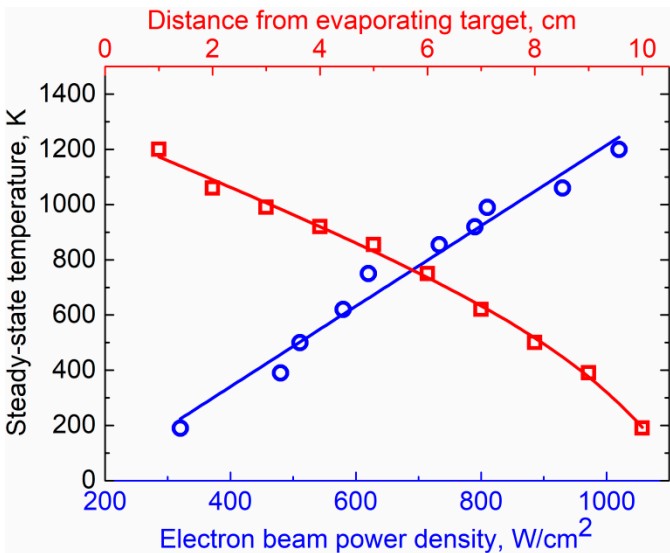

**Figure 3.** Steady-state temperature of the chip at a fixed distance of 2 cm from the target vs. electron beam density, and steady-state temperature of the chip vs. the distance from the target at electron beam power density of 970 W/cm$^2$.

## 3. Results

Figure 4 shows a picture of the layers of the obtained ceramic coating on silicon (witness) and the elemental composition of the coating. The total coating thickness was 70 μm, with the thickness of individual layers of alumina nitride ceramics varying in the range 3–5 μm, and alumina ceramics in the range 20–25 μm.

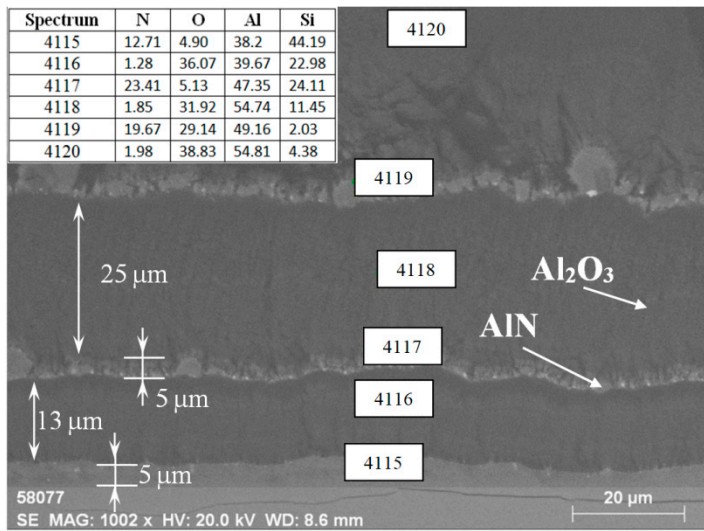

| Spectrum | N | O | Al | Si |
|----------|-------|-------|-------|-------|
| 4115 | 12.71 | 4.90 | 38.2 | 44.19 |
| 4116 | 1.28 | 36.07 | 39.67 | 22.98 |
| 4117 | 23.41 | 5.13 | 47.35 | 24.11 |
| 4118 | 1.85 | 31.92 | 54.74 | 11.45 |
| 4119 | 19.67 | 29.14 | 49.16 | 2.03 |
| 4120 | 1.98 | 38.83 | 54.81 | 4.38 |

**Figure 4.** Micrograph of the silicon transversal section with deposited coatings based on alumina nitride and alumina oxide ceramics and the elemental composition (in weight percent) of the layers (in table inset). Numbers of the spectra acquisition regions are given as the first column in the table inset.

Figure 5 shows, as an example, the pictures of the microchip surface before and after deposition. As seen, despite the somewhat long process of depositing all layers (about 10 min), no visual changes and damages (cracks, metal wires melting, and craters) were observed.

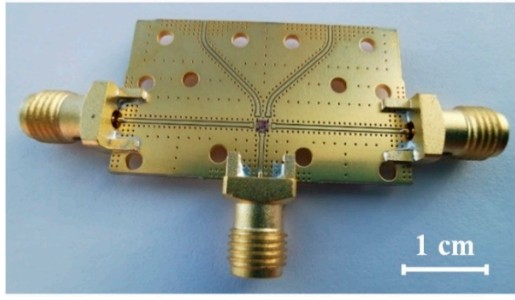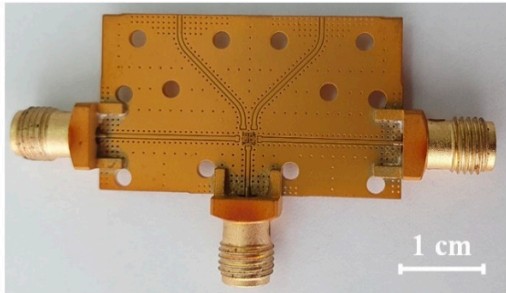

**Figure 5.** The board with Sub-Miniature version A (SMA) plugs before (**left**) and after (**right**) deposition of aluminum oxide and alumina nitride ceramics.

Figure 6 shows micrographs of a microcircuit at all stages of the coating deposition. As seen, there are no obvious defects (except color changes) visible on the surface (micro droplets, cracks, etc.), which confirms the validity of the selected deposition regimes. The total coating thickness was 1.5 µm, with individual layer thickness being 500 nm.

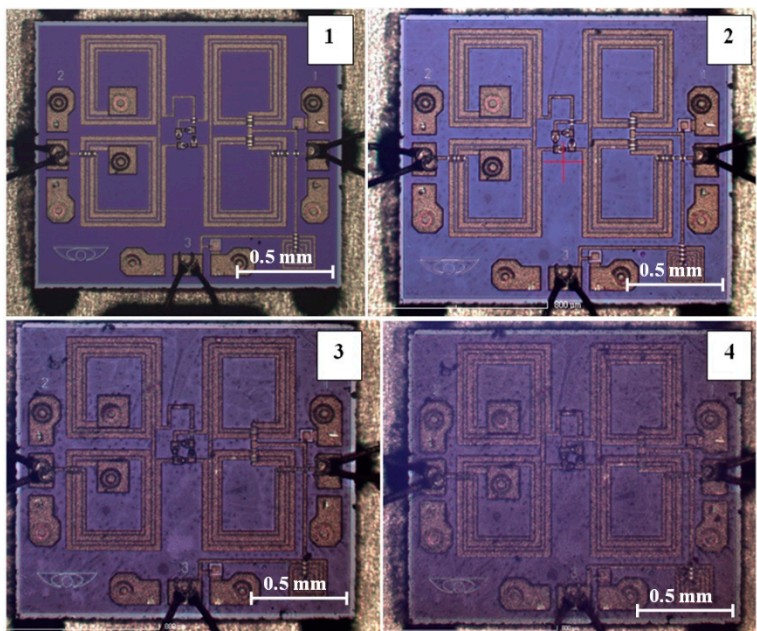

**Figure 6.** Micrograph of the microcircuit at all stages of coating deposition: (**1**)—before deposition; (**2**)—after depositing the first layer ($Al_2O_3$), the coating total thickness is 0.5 µm; (**3**)—after depositing the second layer (AlN), the coating total thickness is 1 µm; (**4**)—after depositing the third layer ($Al_2O_3$), the coating total thickness is 1.5 µm.

Measurement of the temperature growth rate of the chip surfaces during heating without and with coatings of different thickness showed that despite the small value of thickness, the temperature growth rate with increasing thickness decreases, which may speak in favor of using such coatings in devices subject to high temperatures (Figure 7). Investigations of thermal diffusivity and thermal conductivity of the obtained coatings by the "flash" method were carried out using a Discovery Laser Flash DLF-1. This is the most often used method of determining thermal conductivity [12] (Figure 8). The method consists in uniform irradiation of the one plane of a small disk-shaped sample with a laser pulse. The temperature–time dependence on the bottom surface of the disc is recorded by a solid-state optical sensor with an ultra-fast response. Thermal diffusivity and thermal conductivity are determined based on the obtained thermogram taking into account the density of the sample.

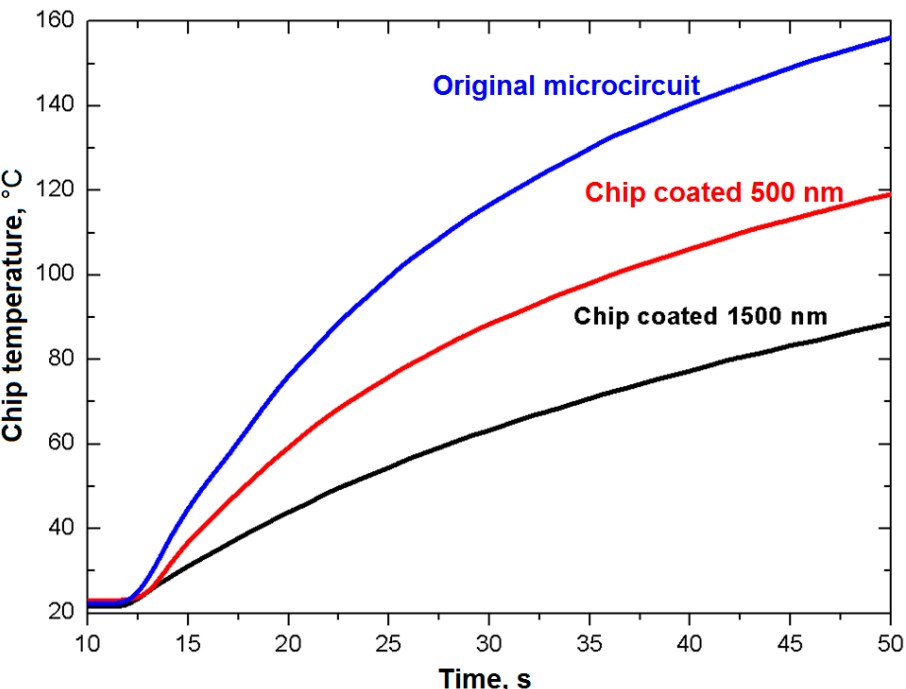

**Figure 7.** The temperature growth rate depending on the chip coating thickness.

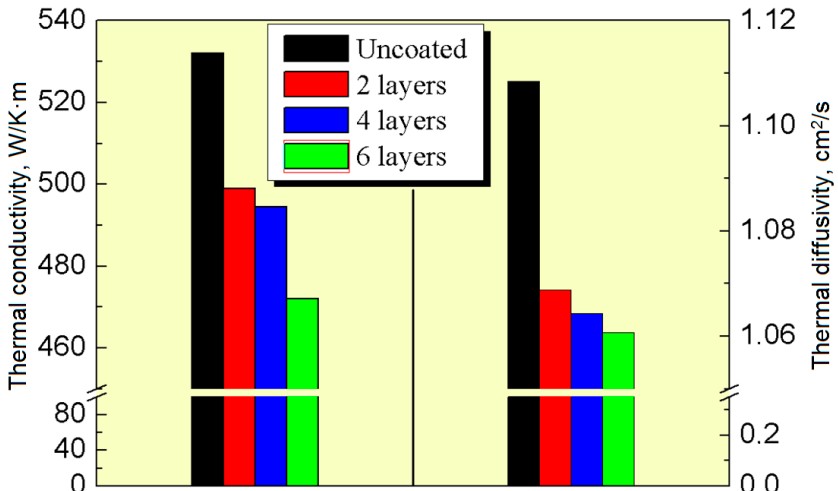

**Figure 8.** Thermal diffusivity and thermal conductivity of the obtained coatings based on aluminum oxide and aluminum nitride ceramics. Each two layers mean the combination of $Al_2O_3$ and AlN.

The mixer measurement regimes: frequency $f_{IF} = 100$ MHz, local oscillator power $P_{LO} = 13$ dBm, signal power $P_{RF} = 0$ dBm. As seen from the experimental data (Figure 7), an increase in the thickness of ceramic-based coating from 500 nm to 1.5 μm does not increase the mixer conversion loss in the high frequency range. Slight performance differences in the lower range of operating frequencies (4–16 GHz) may be caused by instrumental measurement uncertainty. Any significant differences in conversion loss and LO to RF isolation have not been observed (Figure 9).

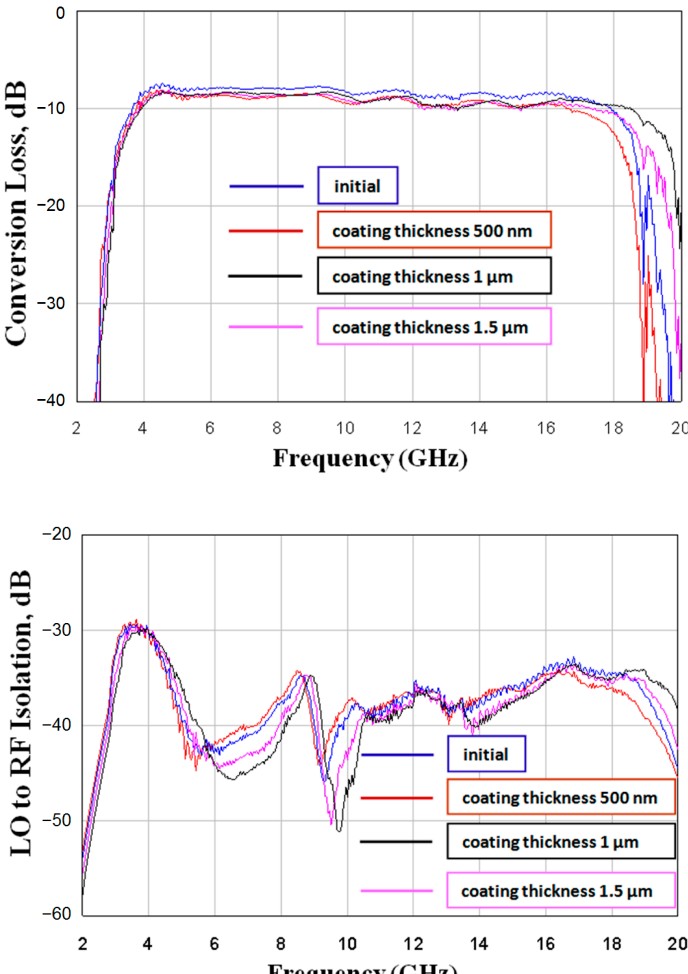

**Figure 9.** The main mixer parameters: conversion loss and LO to RF isolation.

## 4. Conclusions

It is shown that the deposition of AlN- and $Al_2O_3$-based ceramic coatings by a plasma electron source makes it possible to use this technology for MICs with active areas passivated by $Si_3N_4$ dielectric, and apply AlN- and $Al_2O_3$-based coatings in microelectronics to protect against mechanical damage and improve the heat drainage from the surfaces of integrated circuits.

**Author Contributions:** Conceptualization, Y.G.Y. and A.Y.Y.; methodology, Y.G.Y. and A.V.T.; validation, D.B.Z.; investigation, Y.G.Y., A.V.T. and D.B.Z.; resources, Y.G.Y.; writing—original draft preparation, Y.G.Y.; writing—review and editing, D.B.Z.; visualization, A.V.T.; supervision, E.M.O.; project administration, E.M.O.; funding acquisition, E.M.O. All authors have read and agreed to the published version of the manuscript.

**Funding:** This research was funded by the Russian Foundation for Basic Research (RFBR), grant number 18-29-11011 MK.

**Institutional Review Board Statement:** Not applicable.

**Informed Consent Statement:** Not applicable.

**Data Availability Statement:** The data that support the findings of this study are available from the corresponding author upon request.

**Conflicts of Interest:** The authors declare no conflict of interest.

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
