# Peer review of "Electron-Beam Deposition of Aluminum Nitride and Oxide Ceramic Coatings for Microelectronic Devices"

_coatings, doi:10.3390/coatings11060645_

Round 1

Reviewer 1 Report

The authors have conducted a study about the preparation of aluminum oxide and nitride coatings by the method of electron-beam deposition. The purpose and necessity of this study are not well defined. And the content of the article is not qualified for an international journal. The article is more like a short report rather than a scientific article.

Therefore, the article as it stands should be rejected.

Some comments:

(1) P1L10: “nitride and aluminum oxide”?

(2) P3L81: What’s the meaning of “MDM”? Please clarify this term in the manuscript; The same problem for “LO” and “RF” (P6L147);

(3) Materials and Methods: Please explain the sequence of the coating target clearly;

(4) P3L98-99: “To avoid the destruction of the MIC sample, we experimentally selected the optimal temperature for coating deposition”. Which is the optimal temperature and how to know that? Any experiments to support it?

(5) For Fig. 4, how to verify the coating thickness range? Please show it on the SEM image. Also, any EDS mapping to demonstrate the phases of aluminum oxide and nitride?

(6) For Fig. 5, the only color difference on the surface can be detected, and it is pretty hard to show that there is no damage on the surface;

(7) For Fig. 6, any scale bar? It is also hard to conclude that no damage is detected unless zoom in the surface;

(8) For Fig. 7, the chip temperature decreased with increasing the coating thickness, which means the coating materials are not good thermoconducting. Then why the paper title is “Electron-beam deposition of thermoconducting ceramic coatings for microelectronic devices”. It makes readers confused.

Reviewer 2 Report

Manuscript title: Electron-beam deposition of thermoconducting ceramic coatings for microelectronic devices

Authors:  Yury Yushkov, Efim Oks, Andrey Tyunkov, Alexey Yushenko and Denis Zolotukhin

Manuscript No: coatings-1243779

The authors of the manuscript present deposition of protective Al2O3 and AlN ceramics layers on an electronic device using electron beam deposition at low vacuum. The study is of interest, however, there are still some issues to be clarified by the authors. Some figures need to be improved.

  1. Some used abbreviations are not explained in the text: e.g. SMA, LO, RF.
  2. Lines 66-67: maximum beam power density could reach up to 50 W/cm^2 but few lines below text says (Line 72) that “the electron beam power density was no more than 1 kW/cm2”. It seems like a contradiction.
  3. Lines 92-93: “In order to determine the thickness and mechanical properties of the coatings obtained, two “witnesses” of silicon and titanium were positioned near the chip.” It is not clear what the authors mean. In addition, no mechanical properties of the coatings were shown in the manuscript.
  4. Figure 4: green numbers in the figure are not readable. The chemical composition (inset of the figure) is given in atomic or weight %? How was the chemical composition obtained (not described in methods)?
  5. Line 117: text says that no visual changes were visible but figure 5 shows a change in color.
  6. Figures 5,6 scale bars are missing.
  7. Figure 7: Why a wider time interval is not shown? When reaching 50 s, temperature is still increasing.
  8. Figure 8 shows thermal diffusivity and thermal conductivity but there no description in the manuscript of how diffusivity and thermal conductivity were obtained.
  9. Figure 8: Thermal conductivity is decreased after depositing the ceramics layers. Then, the manuscript title may be misleading when describing the layers as thermo-conducting.

Reviewer 3 Report

Yushkov et al. have demonstrated the deposition of AlN and Aluminum oxide in  the chips. Authors have also studied t he thermal  loss based on micro-electronics. I would recommend its publication with minor revision.

1) to deposit aluminum oxide, how did the authors manage arcing effect? Also, thicknesses of films are micron size e-beam is inconvenient. I would like to ask authors opinion which factors did help to obtain micron size films?

Reviewer 4 Report

This manuscript from Yushkov et al. shows their investigation of the e-beam deposited ceramic coatings for thermal conducting applications in microelectronic devices. This method is an interesting and useful technique for improving the heat sinking from the microelectronic elements in modern electronics. In this paper, the authors created the ceramic coatings (AlN and Al2O3) by the e-beam deposition method and showed their study of the thin films, such as morphology, and the thermal properties. They also show the test results of the temperature growth rate for different chip coating thicknesses which demonstrates the potential application of this method in the heat protections of microelectronics. 

In summary, I suggest publishing this paper after minor revisions in both figures and texts to improve readability, as shown in the letter to the authors below.

1) Line 11, the authors mentioned “the beam and plasma methods”, what does the beam means here? Is it the e-beam? I suggest the authors add the full name here to avoid any confusion.

2) In Line 36, the authors mentioned the “fore-vacuum pressure” for the gas, can authors clearly specify the pressure range?   

3) Line 58-59, authors mentioned the heat-conductive AlN layer can distribute the penetrated heat throughout the chip surface, can authors add the reference here?

4) Figure 3, electron beam power is the reason for the steady-state temperature, so I suggest authors consider using the e-beam power as the x-axis and the steady-state temperature as the y-axis.

5) Figure 4, what is the unit of the data in the table? If the 1st column is not useful to the readers, please consider removing it to avoid confusion. 

The two arrows are pointing to the same layer, please consider adjusting the arrows. 

The annotation “Al2O” could be a typo here, please double-check. 

6) Figure 6 and line 124, what is the “individual layer”? Does it mean the individual layer includes two layers: the Al2O3 and AlN layers? 

7) Figure 8, do the 2, 4, 6 layers mean the 1, 2, 3 combined layers? It could be more clear to make the layer number format consistent to figure 6. 

Round 2

Reviewer 1 Report

Thanks for the revision based on the comments.